# Cuban Policosanol (Raydel^®^) Potently Protects the Liver, Ovary, and Testis with an Improvement in Dyslipidemia in Hyperlipidemic Zebrafish: A Comparative Study with Three Chinese Policosanols

**DOI:** 10.3390/molecules28186609

**Published:** 2023-09-14

**Authors:** Kyung-Hyun Cho, Ji-Eun Kim, Seung Hee Baek

**Affiliations:** Raydel Research Institute, Medical Innovation Complex, Daegu 41061, Republic of Korea; ths01035@raydel.co.kr (J.-E.K.); shbaek@raydel.co.kr (S.H.B.)

**Keywords:** policosanol, high-cholesterol diet, high-density lipoproteins, apoA-I, inflammation, interleukin-6, liver, testis, ovary

## Abstract

Many policosanols from different sources, such as sugar cane and rice bran, have been marketed worldwide to improve blood lipid profiles. But so far, no comparative study has commenced elucidating the effect of different policosanols to improve the blood lipid profile and other beneficial effects. This study compared the efficacy of four different policosanols, including one sugar cane wax alcohol from Cuba (Raydel^®^) and three policosanols from China (Xi’an Natural sugar cane, Xi’an Realin sugar cane, and Shaanxi rice bran), to treat dyslipidemia in hyperlipidemic zebrafish. After 12 weeks of consumption of each policosanol (final 0.1% in diet, wt/wt) and a high-cholesterol diet (HCD, final 4%, wt/wt), the Raydel policosanol group and the Xi’an Natural policosanol group showed the highest survivability, of approximately 81%. In contrast, the Xi’an Realin policosanol and the Shaanxi policosanol groups showed 57% and 67% survivability, respectively. Among the five HCD groups, the Raydel policosanol group showed the lowest serum total cholesterol (TC, *p* < 0.001 versus HCD control) and triglyceride (*p* < 0.001 versus HCD control), with the highest percentage of high-density lipoproteins-cholesterol in TC. The Raydel policosanol group also showed the lowest serum aspartate aminotransferase and alanine aminotransferase levels, with the least infiltration of inflammatory cells and interleukin-6 production in hepatocytes with a marked reduction in reactive oxygen species (ROS) production and fatty liver changes. In the ovary, the Raydel policosanol group also showed the highest content of mature vitellogenic oocytes with the lowest production of reactive oxygen species and cellular apoptosis in ovarian cells. In the testes, the Raydel policosanol group also showed the healthiest morphology for spermatogenesis, with the lowest interstitial area and reactive oxygen species production in testicular cells. Conclusively, among the tested policosanols, Cuba (Raydel^®^) policosanol exhibited a comparatively better effect in maintaining zebrafish body weight, survivability, blood lipid profile, hepatic function biomarkers, fatty liver changes, ROS generation, inflammation, and restoration of the cell morphology in ovaries and testes affected by the HCD consumption.

## 1. Introduction

An elevated high-density lipoproteins-cholesterol (HDL-C) quantity and improved quality have been associated with healthy human longevity [1,2] through enhanced antioxidant and anti-inflammatory activities in the blood [3,4]. Dyslipidemia, particularly low levels of serum HDL-C, is a major risk factor for metabolic syndrome, a combination of hypertension, diabetes, and dementia. Dyslipidemia and diabetes are also frequently linked with glycation and oxidative stress, chronic inflammation, and sexual dysfunction [5,6]. Specifically, erectile dysfunction [7] and unexplained female infertility [8] have shown strong associations with dyslipidemia (precisely through low HDL-C).

Many pharmaceuticals and functional foods, such as policosanol, have been developed and marketed to increase HDL-C levels [9,10]. Policosanol is an amalgamation of aliphatic alcohols varying from 24 to 34 carbon atoms, including octacosanol, triacontanol, dotriacontanol, hexacosanol, and tetratriacontanol as the primary constituents, which were extracted from sugar cane (*Saccharum officinarum* L.) wax [11,12] or numerous plants, such as oats and barley, insects, and beeswax [13]. Many policosanols have been purified from multiple plant sources, like sugar cane, rice bran [14,15], wheat germ, and barley sprouts. Despite the diversity of policosanols, among the various sources and origins no reports compare their efficacy in treating dyslipidemia and inflammation in hyperlipidemic animal models.

Cuban policosanol (Raydel^®^) has been used to treat blood dyslipidemia, hypercholesterolemia [16], hypertension [17], gastric cancer [18], and Alzheimer’s disease [19] by increasing the HDL-C and lowering the LDL-C with the removal of amyloid plaque in the brain [19]. In the preceding paper [20], a reconstituted HDL containing Cuban policosanol exhibited the largest particle size and the highest antioxidant, anti-glycation, and anti-inflammatory activity compared to Chinese policosanols [20]. Notably, except for the Cuban policosanol, there is insufficient information about the physiological effects of policosanol on the lipid profile and the extent of inflammation in hyperlipidemic animal models. Furthermore, no information is available on the hidden toxicities and influences on the hepato-endocrinological system and reproduction system by policosanol consumption under hyperlipidemia.

A zebrafish (*Danio rerio*) manifested the induction of hypercholesterolemia by a high-cholesterol diet (HCD) supplementation, and was thus considered an excellent model that mimics dyslipidemia and hepatic inflammation [21,22]. Moreover, adult zebrafish ingesting the HCD for many weeks exhibited an induction of atherosclerosis symptoms, including hypercholesterolemia, lipoprotein oxidation, and fatty streak creation [23]. Hypercholesterolemic zebrafish are a useful experimental model to display the efficiency of drugs or nutraceuticals (such as Cuban policosanol) against liver damage, the infiltration of neutrophils in hepatocytes, and fatty liver change [24]. Hyperlipidemia and oxidative stress are frequently associated with the impairment of the male reproduction system [25] and with female polycystic ovarian syndrome [26]. Also, dyslipidemia and hepatic damage have been linked with the impairment of the semen parameters with abrupt changes in sperm motility [27] and with the incidence of premature ovarian failure [28]. However, information about histological changes of the testis and ovary in hyperlipidemic zebrafish has been scarce.

In view of the above, the present study was commenced to compare the in vivo efficacy of various policosanols obtained from different sources and origins to improve dyslipidemia, hepatic inflammation, and prevention in reproductive organs of zebrafish impaired by HCD supplementation.

## 2. Results

### 2.1. Change in Survivability and Body Weight

After 12 weeks of consumption, the normal diet (ND) group showed 97% survivability. In contrast, the HCD group (final 4%, wt/wt) showed 61% survivability, suggesting that 4% cholesterol supplementation induced acute death via the progression of hyperlipidemia and hyperinflammation (Figure 1A). On the other hand, the PCO1 (Raydel) group and PCO2 (Xi’an Natural) group showed an improved survivability of approximately 81%. In comparison, the PCO3 (Xi’an Realin) and PCO4 (Shaanxi) groups demonstrated lower survivability results of 57% and 67%, respectively.

As shown in Figure 1B, the ND and HCD alone groups showed a 1.8-fold and 2.3-fold enhanced body weight (BW) after 12 weeks of consumption compared with the baseline value (week 0). Under HCD, the PCO1 group and PCO4 group showed a 1.9-fold and 1.8-fold enhancement in BW, while the PCO2 and PCO3 groups showed a 2.3-fold enhancement in BW. These results suggest that the Raydel policosanol group (PCO1) showed the highest survivability and the second lowest increase in BW during the 12 weeks of intake of the HCD-supplemented diet. Interestingly, the PCO4 group showed the lowest enhancement in BW, with lower survivability than the PCO1 group, suggesting that lower body weight was not associated with higher survivability.

### 2.2. Change in Blood Lipid Levels

After 12 weeks, the HCD control group showed the highest total cholesterol (TC) (Figure 2A) and triglyceride (TG) (Figure 2B), 794 ± 18 mg/dL and 889 ± 46 mg/dL, respectively, while the ND group showed 341 ± 12 mg/dL and 244 ± 10 mg/dL, respectively, suggesting that 4% cholesterol supplementation enhanced 2.3-fold and 3.6-fold blood TC and TG levels compared to the ND groups. On the other hand, the PCO1 group showed the lowest TC (425 ± 7 mg/dL) and TG (518 ± 15 mg/dL) levels, which were up to 47% and 32% lower than the HCD alone group, respectively, while PCO3 and PCO2 groups showed 32% and 30% lower TC, respectively, than the HCD alone group. The results signify that the Chinese policosanol groups had weaker blood lipid lowering profiles (TC and TG) than the Cuban policosanol (PCO1).

As shown in Figure 2C, the HCD group (253 ± 6 mg/dL) showed higher HDL-C levels than the ND group (217 ± 4 mg/dL); however, the HCD group showed 50% lower HDL-C/TC (%) than the ND group (Figure 2D). The PCO1 group showed a lower serum HDL-C level (299 ± 6 mg/dL) (Figure 2C) than that of PCO2 group. The HDL-C/TC (%) was highest (68%) in the PCO1 group. Compared to only the HDL group, PCO2, PCO3, and PCO4 groups showed an adequate increase in HDL-C/TC (%), quantified as 53%, 55%, and 41%, respectively.

As shown in Appendix A, the LDL-C level was elevated remarkably in the HCD group (406 ± 24 mg/dL), while the ND group showed 34 ± 3 mg/dL, suggesting that 4% cholesterol supplementation elevated the LDL-C level. Interestingly, the PCO1 group showed the lowest LDL-C (32 ± 2 mg/dL), which was nearly similar to the value obtained for the ND group (34 ± 3 mg/dL). The other policosanol groups (PCO2, PCO3 and PCO4) also showed an adequate effect to lower the HCD-elevated LDL-C levels. The LDL-C levels in the PCO2, PCO3, and PCO4 groups were 187, 90, and 178 mg/dL, correspondingly.

The LDL-C: HDL-C (L:H) ratio was elevated in the HCD control group (Appendix A) that was efficiently reduced by PCOs. However, the most prominent effect was observed in the PCO1 group (Appendix A).

A calculation of the total amount of non-HDL-C showed that the HCD control group had a 4.4-fold higher level than the ND control group, suggesting that the cholesterol supplementation caused the elevation of non-HDL-C (Appendix A). All the PCOs groups significantly (*p* < 0.05) reduced the HCD-elevated non-HDL-C level, though the most effective results were observed for the PCO1 group (136 ± 5 mg/dL) where non-HDL-C were found to be the least among all the tested groups, including the ND group. The Cuban policosanol group (PCO1) showed a remarkable 75% reduction in non-HDL-C compared to the HCD group.

The ratio of TG/HDL-C was significantly elevated by HCD, which was reverted significantly (*p* < 0.05) in PCO1 and PCO2 groups. In contrast to this, PCO3 and PCO4 groups did not show any significant (*p* < 0.05) effects on the reduction in TG/HDL-C elevated by HCD supplementation (Appendix A).

These results collectively suggest that Cuban policosanol (Raydel^®^) consumption (PCO1 group) resulted in the highest HDL-C content with respect to TC along with the lowest L:H ratio and TG:HDL-C ratio. In contrast, the other Chinese policosanols showed weaker effects in increasing the HDL-C and HDL-C/TC (%).

### 2.3. Change in Hepatic Function Parameters

After 12 weeks of consumption, the HCD control group showed 1.6-fold (*p* = 0.024) and 1.5-fold (*p* = 0.044) higher blood AST and ALT levels than those of the ND control group (Figure 3), suggesting that cholesterol (final 4%) supplementation-associated damage occurred in the liver and muscle. The PCO1 and PCO2 groups displayed 18% and 24% reduction in the HCD-elevated AST level. However, PCO3 and PCO4 groups did not display an impact on the HCD-elevated AST level (Figure 3A).

Similar to the AST results, the HCD-elevated ALT level was significantly decreased by 45% (*p* = 0.010) in the PCO1 group, as compared to the HCD control group (Figure 3B). The PCO2, PCO3, and PCO4 groups showed 27% (*p* = 0.026), 18%, and 23% lower levels than the HCD control, respectively. Although all policosanol groups showed lower ALT levels than the HCD control group, the extent of the serum ALT levels differed according to the origins, suggesting that the hepatic protection activity of policosanol may vary depending on the composition of long-chain aliphatic alcohols. Cuban policosanol showed remarkably higher protective activity against hepatic damage from high cholesterol consumption, whereas Chinese policosanols, particularly Xi’an Realin and Shaanxi, did not show protection activity.

### 2.4. Analysis of the Liver Tissue (H&E Staining)

Hepatic tissue analysis employing H&E staining was performed to observe the hepatological changes in different groups, and the results are depicted in Figure 4. The HCD control group (photograph b) exhibited the highest H&E-stained area (Figure 4A), with a massive neutrophil infiltration that was significantly 10.9-fold higher (*p* = 0.008) than the ND control group (photograph a). In contrast, the PCO1 group (photograph c) significantly inhibited the HCD-induced neutrophil efflux, evident through ~72% lower (*p* = 0.005) neutrophil counts than the HCD control group (Figure 4B). In contrast to this, the other policosanol groups (PCO2, PCO3 and PCO4) did not have much impact on alleviating the neutrophil infiltration provoked by HCD (Figure 4B). The H&E-stained area observed for PCO2, PCO3, and PCO4 groups showed similarity with the HCD control group (photographs d, e, f) (Figure 4). PCO2, PCO3, and PCO4 groups showed non-significant 7–13% lower neutrophil counts compared to the HCD control group.

### 2.5. Fatty Liver Changes and ROS Production

Oil red O staining with hepatic section showed that the HCD control group (photo b1) had an 8.1-fold higher oil red stained area than the ND group (photo a1), as shown in Figure 5A, implying a severe fatty liver change in response to cholesterol supplementation. Under HCD consumption, the PCO1 group (photo c1) and PCO2 group (photo d1) efficiently prevented the fatty liver changes with a marked 75% and 73% lower oil red stained area, respectively, than the HCD control group (Figure 5B). On the other hand, the PCO3 group (photo e1) and PCO4 group (photo f1) showed no effect against HCD-induced fatty liver changes. Moreover, PCO3 and PCO4 groups displayed a 14% and 26% higher oil red stained area than the HCD control group.

DHE staining revealed that the HCD control group (photo b2) had 3.0-fold higher ROS production than the ND group (photo a2), signifying the impact of cholesterol consumption on ROS production. For the HCD supplementation, PCO1 (photo c2), PCO2 (photo d2), and PCO3 (photo e2) groups showed 68%, 58%, and 39% reduced ROS production, respectively, compared to the HCD control group, quantified by measuring the DHE stained area (Figure 5B). Contrary to this, the PCO4 group (photo f1) had no inhibitory effect against HCD-induced ROS production. Moreover, 28% higher ROS production was observed in the PCO4 group compared to the HCD control group, indicating severe ROS production and fatty liver changes.

### 2.6. IL-6 Production in Liver Tissue (Immunohistochemistry)

As shown in Figure 6A, the immunohistochemistry (IHC) of hepatic tissue revealed higher IL-6 production in the HCD control group (photo b), evident by a 7.8 times higher (*p* < 0.001) IHC stained area (Figure 6B) than that of the ND control group (photo a), indicating the impact of cholesterol supplementation for 12 weeks on IL-6 accumulation in hepatic tissue. The PCO1 group (photo c) showed the least IL-6-stained area (3.6 ± 0.3%), which was 81% lower (*p* < 0.001) than the IHC-stained area of the HCD control, signifying the impact of PCO1 to alleviate the HCD-induced IL-6 production. Likewise, the PCO2 group (photo d) also lowered the IL-6 production, with a marked 59% lower (*p* = 0.008) IL-6-stained area than that of the HCD control. In contrast, the PCO3 and PCO4 groups did not reduce IL-6 production induced by HCD. Indeed, a 21.7 ± 8.7% and 29.2 ± 4.4% higher IL-6 stained area, that is, 1.2-fold (*p* = 0.006) and 1.5-fold higher (*p* < 0.001) than the IHC stained area of HCD control, was observed in PCO3 and PCO4 groups, respectively (Figure 6B). The results suggest the higher efficacy of PCO1 and PCO2 to counter inflammation and consequent effects on serum AST/ALT levels, fatty liver changes, and ROS production in the liver.

### 2.7. Examination of Ovarian Tissue Section

H&E staining (Figure 7A,B) disclosed that the HCD group had the highest previtellogenic oocyte content (~83%) and the lowest early and mature vitellogenic oocyte content (~13% and 3%, respectively), compared to ND group. On the other hand, the PCO1 group showed the highest early and mature vitellogenic oocyte content (~47%), with the lowest premature oocyte content (~53%). In contrast, the three Chinese policosanols (PCO2, PCO3 and PCO4) showed a much higher premature oocyte content (~74–80%) with a lower early and mature vitellogenic oocyte content (~20–26%).

DHE staining showed that the HCD group had a significantly 2.9-fold stronger red fluorescent intensity than the ND group, indicating higher ROS production in oocytes in response to HCD consumption (Figure 7A,C). The PCO1 group showed the least red fluorescent intensity, which was 85% lower than the fluorescent intensity quantified in the HCD group, signifying the impact of PCO1 against HCD-induced ROS production. Likewise, the PCO2 group showed a 33% lower DHE-stained area than the HCD control group. However, PCO3 and PCO4 groups showed no impact against HCD-induced ROS production, evidenced by the similar DHE stained area observed for the HCD control group (Figure 7C).

AO staining revealed a 23-fold higher green intensity in the HCD group than in the ND group (Figure 7A,C), suggesting the association of high cholesterol consumption with increased cellular apoptosis. The PCO1 and PCO2 groups showed 93% and 75% lower green fluorescent intensity corresponding to the degree of apoptosis than the HCD control group, respectively. In contrast, the PCO4 groups showed a 35% higher AO-stained area than the HCD control group, suggesting the greatest extent of cellular apoptosis in the oocytes of the PCO4 group. These results combined suggest that PCO1 consumption resulted in the highest mature oocyte content with the lowest extent of cellular apoptosis and ROS production in ovarian cells.

### 2.8. Analysis of Testis Tissue Section

The H&E staining (Figure 8A) analysis of the testis section showed that the ND group (photo a) had healthy seminiferous tubules with a full cell population adherent to the basal membranes without any notable gaps between the membranes and interstitium, approximately 19% of interstitial area (Figure 8B). In contrast to this, the HCD control group (photo b) displayed irregularly outlined seminiferous tubules with disarranged cellular layers as well as a broken lamina basal membrane with the largest interstitial area of approximately 36%, indicating an impaired ability of spermatogenesis. The morphometric results of spermatogenesis in the HCD group (photo b) showed that spermatids and sperm (mature forms) were smaller than in the ND group (photo a). In contrast, the PCO1 group showed a meticulous increase in the area of spermatids, as shown in Figure 8A (photo c), with the smallest interstitial area among HCD groups. The average interstitial areas of the PCO1 and PCO2 groups were approximately 26% and 32% smaller than the HCD control group, respectively (*p* < 0.001). In contrast, the PCO3 and PCO4 groups showed 39% and 37% of the average interstitial areas, respectively, which was higher than that of the average interstitial areas in the HCD control group (Figure 8B).

DHE staining showed that the HCD control group had a 19% red-stained area, indicating 3-fold higher ROS production than the ND control group (~6% red-stained area) (Figure 9A,B). The PCO1 group showed the lowest ROS production (~3% red-stained area), followed by the PCO2 group (~12% red-stained area) (Figure 9B). On the other hand, the PCO3 and PCO4 groups did not affect HCD-induced ROS production.

AO staining showed that the HCD group had 33-fold higher green intensity than the ND group (Figure 9), suggesting that high cholesterol consumption caused a remarkable increase in cellular apoptosis in the testis. The PCO1 and PCO2 groups showed 93% and 75% lower green intensity, respectively, than the HCD control group. In contrast, the AO staining of the PCO3 and PCO4 groups displayed a non-significant effect of PCO3 and PCO4 on the inhibition of HCD-induced apoptosis. Combined histologic results suggest that the PCO1 group had the highest protective activity with the least extent of cellular apoptosis and ROS production in spermatogenesis.

## 3. Discussion

The current study was orchestrated to differentiate the in vivo efficacy of various policosanols on the blood lipid-lowering effect, anti-inflammatory activity, and protection of liver, ovary, and testis tissue from oxidative damage in hyperlipidemic zebrafish. The results showed that 12 weeks of Cuban policosanol (Raydel^®^) (PCO1) utilization resulted in the highest survivability (Figure 1A) and potent efficacy to lower the blood TC, LDL-C, and TG with higher HDL-C/TC (%) under HCD supplementation (Figure 2 and Appendix A). In liver function, the Cuban policosanol group (PCO1) showed the lowest blood AST and ALT level (Figure 3) with the least infiltration of neutrophils in hepatocytes (Figure 4). Histology analysis of the liver revealed the PCO1 group to have the least fatty liver change and ROS production (Figure 5) with the least IL-6 production (Figure 6). In ovarian tissue, the PCO1 group showed the highest percentage of mature vitellogenic-stage oocytes with the least ROS production and cellular apoptosis (Figure 7). In the testis, the PCO1 group showed the least damage to the spermatogenesis cell morphology with the smallest interstitial area (Figure 8), ROS production, and apoptosis (Figure 9).

The results showed that hyperlipidemia, which was induced by a high-cholesterol diet (final 4%), was associated with affliction to the hepatic function, the elevation of ROS production and inflammation in the liver (Figure 3, Figure 4, Figure 5 and Figure 6), and the impairment of male and female reproductive organs, such as the testis and ovary (Figure 7, Figure 8 and Figure 9). The HCD control group showed the highest serum AST and ALT levels with severe fatty liver changes and ROS production after 12 weeks of consumption, consistent with previous reports suggesting Cuban policosanol’s impact on hepatic biomarkers and ROS production [24]. The HCD control group showed a remarkable increase in IL-6 production in the liver, a severe decrease in mature vitellogenic oocytes, and the enlargement of the interstitial area in the testis. These results suggest that dyslipidemia is associated with hepatic inflammation and infertility, which is in good agreement with previous reports showing that mice consuming a high-cholesterol diet for 12 weeks showed an impairment in sperm maturation and capacitation [29].

In contrast, the co-supplementation of Cuban policosanol in the HCD (final 0.1%, wt/wt) improved the lipid profile and liver functions. These results concurred with previous clinical studies documenting that twelve weeks of Cuban policosanol (Raydel^®^) consumption offered protection of the liver functions via lowered AST, ALT, γ-GTP, and alkaline phosphatase (ALP) levels in Japanese participants [30,31]. In addition, the serum antioxidant abilities were elevated by policosanol consumption at week 12, with a 37% increase in ferric ion reduction ability (FRA) and 29% increase in paraoxonase (PON) activity [30]. Interestingly, the purified HDL_2_ and HDL_3_ from the policosanol group also showed enhanced FRA and PON ability, approximately 1.3–1.6-fold higher than the placebo group [31]. Similarly, 12 weeks of Cuban policosanol consumption (5 mg and 10 mg per day) by hypercholesterolemic patients resulted in a significant decrease in ALT and γ-GTP, a decrease in TC and LDL-C, and an increase in HDL-C [32]. Furthermore, the in vitro comparison of antioxidant ability showed that Cuban policosanol had the highest potent inhibition ability of LDL oxidation in the presence of cupric ions [20]. Under the in vitro treatment of fructose, Cuban policosanol showed the highest inhibition activity of the extent of glycation with the prevention of apoA-I degradation [20]. Furthermore, the clinical study with healthy Japanese participants showed that the policosanol group had 4% lower glycated hemoglobin (Hb_A1c_) levels than those at week 0 and the placebo group at week 12 [30,31]. These enhanced serum antioxidant abilities and hepatic function parameters might help improve the hepatic function and reproductive organs, testis, and ovary.

It has been well-established that oxidative stress promotes inflammation in the ovaries and causes ovarian aging and infertility [33]. The consumption of a high-cholesterol diet for 21 weeks in zebrafish caused a remarkable increase in AST and ALT with an impairment of the testicular morphology; the HCD group showed a 1.7-fold greater enlargement of the interstitial area than the ND group [34]. To the best of the authors’ knowledge, the current study is the first to show that Cuban policosanol (Raydel^®^) consumption could alleviate damage to the liver and reproduction system, ovary, and testis under high cholesterol consumption. The alleviation of hepatic inflammation and ROS production is based on the increase in HDL-C and the enhancement of HDL functionality, as shown in previous reports with hyperlipidemic zebrafish [24] and spontaneously hypertensive rats (SHR) [35]. In hyperlipidemic zebrafish, eight weeks of policosanol consumption ameliorated the elevated infiltration of neutrophils, ROS production, and fatty liver changes with a significant reduction in AST and ALT [34]. In the SHR, elevated oxidized species in hepatic tissue and C-reactive protein in the blood were reduced by eight weeks of policosanol consumption in a dosage-dependent manner [35].

A previous study reported that policosanol did not impair the male reproduction system without genetic defects from long-term supplementation (5, 50, and 500 mg/kg of body weight/day) throughout three successive generations [36]. For female reproduction ability, there were no teratogenic effects in the rats and rabbits given 500–1000 mg/kg of body weight/day two weeks before mating and throughout mating and pregnancy to day 21 of lactation [37]. Although no reports describe the improvement in the reproduction ability by policosanol consumption, these results showed that the treatment of dyslipidemia with Cuban policosanol is associated with improved ovarian cell and testicular cell morphology and functions. Moreover, it has been reported that a six-week consumption of octacosanol elevated gene expressions of glucose transporter protein (GLUT-4) and adenosine monophosphate protein kinase (AMPK) in muscle and liver tissues of weaning piglets [38]. Increased oxidative stress in hyperlipidemia has been associated with fatty liver change and pro-inflammatory responses to cause apoptosis in hepatic tissue [39]. It has been reported that policosanol prevented bone loss and decreased bone resorption in ovariectomized rats [40]. More recently, policosanol activated osteoblast differentiation via AMPK-mediated expression of insulin-induced genes to facilitate zebrafish fin regeneration [41]. These results are in good agreement with the PCO1 group, which showed the lowest apoptosis in the ovary and suggested that Cuban policosanol exhibited potent anti-apoptosis activity in hyperlipidemic zebrafish.

On the other hand, determining the microbiome composition could provide very important information to assess the efficacy of policosanol in hyperlipidemic mice, as reported previously [42]. Policosanol consumption (0.5% in diet wt/wt) increased *Bacteroides* (B) and decreased *Firmicutes* (F), increased the B/F ratio in the hyperlipidemic mice and also increased HDL-C and decreased TC and LDL-C [42]. These results suggest that the treatment of dyslipidemia might be associated with antioxidant, anti-inflammatory, and anti-apoptosis activity via improvements in microbiota constituents. Future study is necessary to compare the change in microbiome compositions depending on the different origins of policosanol using the hyperlipidemic vertebrate model.

The current results showed that hyperlipidemia and fatty liver changes are also linked to severe impairments in the reproductive organs (Figure 7, Figure 8 and Figure 9). HCD supplementation caused a decrease in the mature vitellogenic stage of approximately 2.7%, while the ND group showed 6.5%, indicating that elevated oxidative stress caused by hyperlipidemia inhibited oocyte maturation, as reported elsewhere [43,44]. In the current study, however, the co-supplementation of Cuban policosanol resulted in the largest increase in early-vitellogenic and mature-vitellogenic oocytes, indicating enhanced oocyte maturation and folliculogenesis. These results may apply to develop a nutraceutical with antioxidant and anti-inflammatory activity to simultaneously treat hyperlipidemia, hepatic inflammation, and impairments of the reproduction system in humans, as summarized in Figure 10. Among all the tested policosanols, Cuban policosanol (Raydel^®^) (PCO1) displayed the highest activity for most of the tested biological parameters. Probably, this is due to the abundance of octacosanol in Cuban policosanol, compared to Xi’an Natural (PCO2), Xi’an Reali (PCO3), and Shaanxi (PCO4) policosanols [20]. The statement is validated by earlier reports suggesting the vital role of octacosanol as an antioxidant and anti-inflammatory agent [45]. Furthermore, several studies have documented octacosanol’s protective role in liver disorders [46]. Also, the higher amount of triacontanol and tetracosanol in Cuban policosanol (PCO1) compared to Xi’an Natural (PCO2) and Shaanxi (PCO4) [20] might be an additional reason for the better biological activity of PCO1, which is in accordance with the earlier report deciphering the important bifunctionality triacontanol and tetracosanol [45,46].

## 4. Materials and Methods

### 4.1. Materials

Raydel^®^ sugar cane wax alcohol, policosanol (PCO) 1, was obtained from the National Center for Scientific Research (CNIC), Havana, Cuba. Chinese sugarcane policosanol, PCO2 and PCO3, were purchased from Xi’an Natural Field Biotechnique Co, Ltd. (Xi’an, China) and Xi’an Realin Biotechnology Co, Ltd. (Xi’an, China), respectively. Chinese rice bran policosanol, PCO4, was obtained from Shaanxi Pioneer Biotech (Xi’an, China). All raw materials of each policosanol were examined utilizing the same gas chromatography, as described previously [20].

### 4.2. Zebrafish Maintenance and Policosanol Supplementation

Zebrafish and embryos were maintained per established protocols outlined in the Guide for the Care and Use of Laboratory Animals [47,48]. The procedures for using zebrafish were authorized by the Animal Care and Use Committee at the Raydel Research Institute under approval code RRI-20-003, Daegu, Republic of Korea, in accordance with the ARRIVE guidelines 2.0 [49]. The zebrafish were housed in a controlled environment within a system cage, maintained at 28 °C temperature throughout the experiment with a 12 h light and 12 h dark cycle and provided with a diet of normal tetrabits (TetrabitGmbh D49304, Melle, Germany).

The 10-week-old zebrafish were randomly divided into six groups, as shown in Table 1; each group had 70 zebrafish and were supplemented with one of each policosanol (final 0.1%, wt/wt) with HCD (final 4% cholesterol, wt/wt) in the zebrafish diet (Tetrabit, Gmbh D49304, Melle, Germany) for 12 weeks. The groups were as follows: exposed to normal diet (ND, control, n = 70), exposed to 4% high-cholesterol diet (HCD, n = 70), PCO1 (exposed to Cuban policosanol, Raydel^®^, with HCD, n = 70), PCO2 (exposed to Xi’an Natural policosanol with HCD, n = 70), PCO3 (exposed to Xi’an Realin policosanol with HCD, n = 70), and PCO4 (exposed to Shaanxi policosanol with HCD, n = 70). Before feeding each policosanol, all groups were acclimated to HCD for four weeks except the ND control group. Subsequently, the zebrafish were supplemented with 0.01 mg each of policosanol in 10 mg of Tetrabit and fed twice daily at 9 am and 6 pm with 20 mg of the designated diet, containing 0.02 mg of policosanol/day per zebrafish.

### 4.3. Blood Collection and Analysis

For plasma lipid analysis, blood was collected from zebrafish of different groups after 12 weeks of supplementation, as in a previously adopted method [23,24]. In brief, blood from the zebrafish was collected and immediately mixed with 3 μL PBS comprising 0.3 mg/mL ethylenediaminetetraacetic acid (EDTA) followed by 15 min centrifugation at 5000× *g*. The supernatant was collected and processed for the quantification of total cholesterol (TC) and triglyceride (TG), employing a colorimetric assay kit (Cholesterol, T-CHO, and TG, Cleantech TS-S; Wako Pure Chemical, Osaka, Japan). HDL-C (AM-202), alanine transaminase (ALT) (AM-103K), and aspartate transaminase (AST) (AM-201) were quantified using a commercial detection kit (Asan Pharmaceutical, Hwasung, Republic of Korea), following the suggested methodology prescribed by the manufacturer.

### 4.4. Analysis of Hepatic Tissue

The liver of zebrafish from different groups was retrieved surgically after sacrificing them. The liver tissue was preserved in 10% formalin for 24 h following ethanol dehydration. The dehydrated tissue was embedded in paraffin, followed by 7 μm thick sectioning that was subsequently treated with poly-L-lysine and smeared with hematoxylin and eosin (H&E). The stained tissue was perceived under an optical microscope (Motic microscopy PA53MET, Hong Kong, China), and the Image J platform (http://rsb.info.nih.gov/ij/, accessed on 15 September 2022) was utilized to quantify the number of neutrophils (stained dark violet to blue color). 

The section of the hepatic tissue was sliced using a microtome (Leica, CM1510s, Heidelberg, Germany). The tissue section (7 μm thick) was stained for 10 min with oil red O (Cat#O0625, Sigma, St. Louis, MO, USA), followed by washing with water and subsequent counterstaining with hematoxylin. After 2 min incubation, the stained area was washed with water and visualized under an optical microscope (Motic Microscopy PA53MET, Hong Kong, China).

Dihydroethidium (DHE) fluorescent staining was performed to quantify the ROS production method, as previously described [50]. The 7 μm thick tissue section was stained with DHE (cat # 37291; Sigma, St. Louis, MO, USA) (final 30 μM) for 30 min in the dark. After washing three times with water, the stained section was visualized under a fluorescent microscope (Nikon Eclipse TE2000, Tokyo, Japan), and the fluorescent intensity was quantified using the Image J platform (http://rsb.info.nih.gov/ij/, accessed on 15 September 2022) by converting fluorescent images to type 8 bit, following threshold level at the fixed value of 150. Finally, the fluorescent stained area was measured by subtracting the background intensity.

IL-6 production in the hepatic tissue was quantified by immunohistochemical staining. In brief, a 7 μm thick tissue section was inundated with primary anti-IL-6 antibody (ab9324, Abcam, London, UK). After overnight incubation at 4 °C, the tissue section was developed using Envision + system kits (code 4001, Dako, Denmark) comprising horseradish peroxidase (HRP)-conjugated secondary antibody against the IL-6-specific primary antibody. The IL-6 quantification was achieved by employing Image J software (http://rsb.info.nih.gov/ij/, accessed on 16 May 2023) by converting IL-6-stained images to RGB stack. Following the threshold levels [of lower limit (20) and upper limit (120)] to minimize the inclusion of background staining, all the images were processed at the same threshold values and the resulting % area was accredited as the IL-6-stained area.

### 4.5. Analysis of Ovarian Tissue

The pre-vitellogenic, early-vitellogenic, and mature-vitellogenic stages of the ovary were differentiated manually, as described previously [51,52]. The classification criteria were as follows. For the pre-vitellogenic stage, follicles measuring 250 μm diameter or smaller were categorized as such, including the smallest pre-vitellogenic follicles and those in the perinucleolar stage. Follicles 250–500 μm in diameter were classified as being in the early-vitellogenic stage. This encompassed the largest pre-vitellogenic follicles and those at the peripheral cortical alveolar stage. Follicles 500 μm in diameter or larger were assigned to the mature-vitellogenic stage. This stage included follicles characterized by yolk-filled alveoli distributed throughout the ooplasm and vitellogenic follicles. The ovary follicles were effectively categorized into their respective stages of development using these classification criteria.

### 4.6. Analysis of Testes Tissue

The testes were surgically removed from the zebrafish-fed ND control, HCD control, and HCD + in each policosanol-supplemented group and immersed in Bouin solution for 2 days, followed by 2 days of dehydration in 30% sucrose solution. Finally, the tissue was embedded in paraffin and frozen. The frozen tissue was sectioned using a microtome (Leica, CM1510s, Heidelberg, Germany). The testes section was processed for H&E staining, as in the previously described method [34,53]. The H&E-stained area was visualized under the microscope to examine the spermatogenesis, seminiferous tubules, and interstitial space, as defined previously [34,54]. In brief, interstitial spaces in the H&E-stained images were quantified using Image J software (http://rsb.info.nih.gov/ij/, accessed on 16 May 2023) by converting the images to RGB stack. Following the threshold level [of lower limit (220) and upper limit (255)] obtained percentage values corresponding to the interstitial space. A minimum of at least three zebrafish from each group were examined for the histological analysis of testicular tissue.

### 4.7. Statistical Analysis

The outcomes are illustrated as mean ± SEM for all the experiments conducted in triplicate. Statistical comparisons amidst the groups were performed using a one-way ANOVA in SPSS software (version 28.0; SPSS Inc., Chicago, IL, USA). Post hoc analysis using the Dunnett’s test was evaluated to conclude any significant disparity among the groups at *p* < 0.05.

## 5. Conclusions

Twelve weeks of Cuban policosanol (Raydel^®^) consumption efficiently improved the HCD-altered blood lipid profile, hepatic inflammation, fatty liver changes, ROS production, and cell morphology in the ovary and testis. Also, Cuban policosanol (Raydel^®^) efficiently improved zebrafish survivability and mitigated body weight disruptions caused by HCD consumption. These findings established the different functionality of Cuban policosanol (Raydel^®^) compared to Chinese policosanols regarding in vivo efficacy to prevent hyperlipidemia, hepatic inflammation, and impairments of the reproduction system.

## Figures and Tables

**Figure 1 molecules-28-06609-f001:**
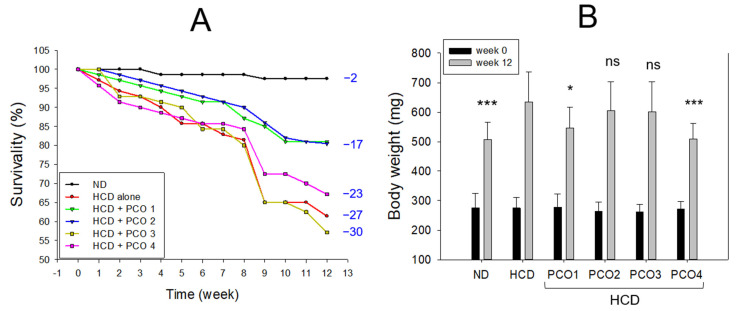
Change in survivability (**A**) and body weight (**B**) during 12 weeks’ supplementation of each policosanol under high-cholesterol diet. HCD, high-cholesterol diet; ND, normal diet; PCO1, Raydel policosanol; PCO2, Xi’an Natural policosanol; PCO3, Xi’an Realin policosanol; PCO4, Shaanxi policosanol. Numerical values in the blue font indicate dead zebrafish numbers during 12 weeks of consumption. Data are expressed as mean ± SEM. Statistical differences in multiple groups were compared using a one-way analysis of variance (ANOVA) with Dunnett’s post hoc test. *, *p* < 0.05 versus HCD control; ***, *p* < 0.001 versus HCD control; ns, not significant versus HCD control.

**Figure 2 molecules-28-06609-f002:**
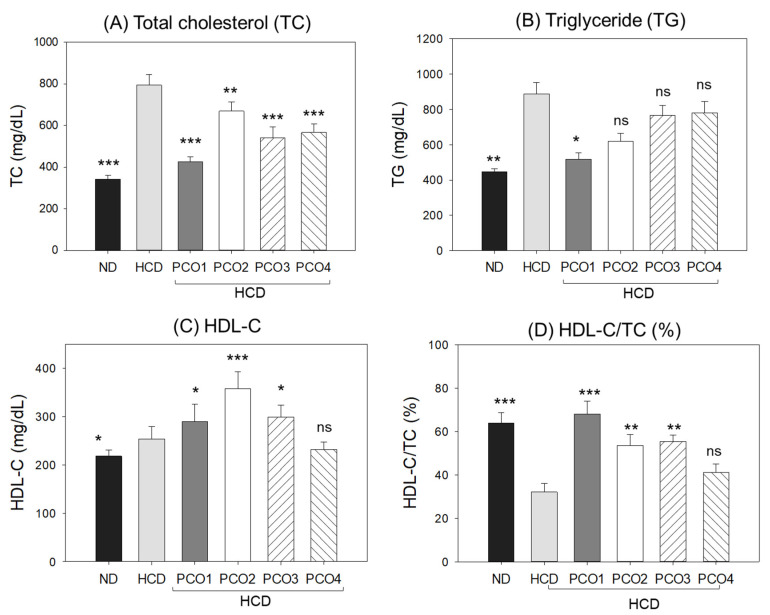
Total cholesterol (TC), triglyceride (TG), high-density lipoproteins cholesterol (HDL-C), and the HDL-C and total cholesterol ratio (HDL-C/TC, %) in the blood after 12 weeks of supplementation of each policosanol (PCO) under high-cholesterol diet consumption. HCD, high-cholesterol diet; ND, normal diet; PCO1, Raydel policosanol; PCO2, Xi’an Natural policosanol; PCO3, Xi’an Realin policosanol; PCO4, Shaanxi policosanol. Data are expressed as mean ± SEM. Statistical differences between multiple groups were compared using a one-way analysis of variance (ANOVA) with Dunnett’s post hoc test between the other group and the HCD group. *, *p* < 0.05 versus HCD control; **, *p* < 0.01 versus HCD control; ***, *p* < 0.001 versus HCD control. ns, not significant versus HCD control.

**Figure 3 molecules-28-06609-f003:**
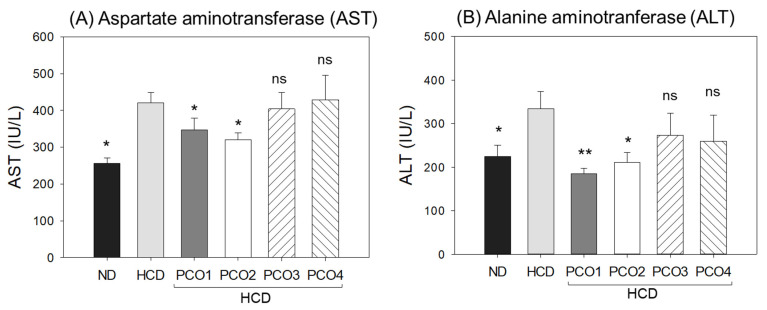
Comparison of the blood hepatic damage parameters of aspartate aminotransferase (AST) and alanine aminotransferase (AST) after 12 weeks supplementation of each policosanol under high-cholesterol diet consumption. HCD, high-cholesterol diet; ND, normal diet; PCO1, Raydel policosanol; PCO2, Xi’an Natural policosanol; PCO3, Xi’an Realin policosanol; PCO4, Shaanxi policosanol. Data are expressed as mean ± SEM. Statistical differences in multiple groups were compared using a one-way analysis of variance (ANOVA) with Dunnett’s post hoc test between the other group and the HCD group. *, *p* < 0.05 versus HCD control; **, *p* < 0.01 versus HCD control; ns, not significant.

**Figure 4 molecules-28-06609-f004:**
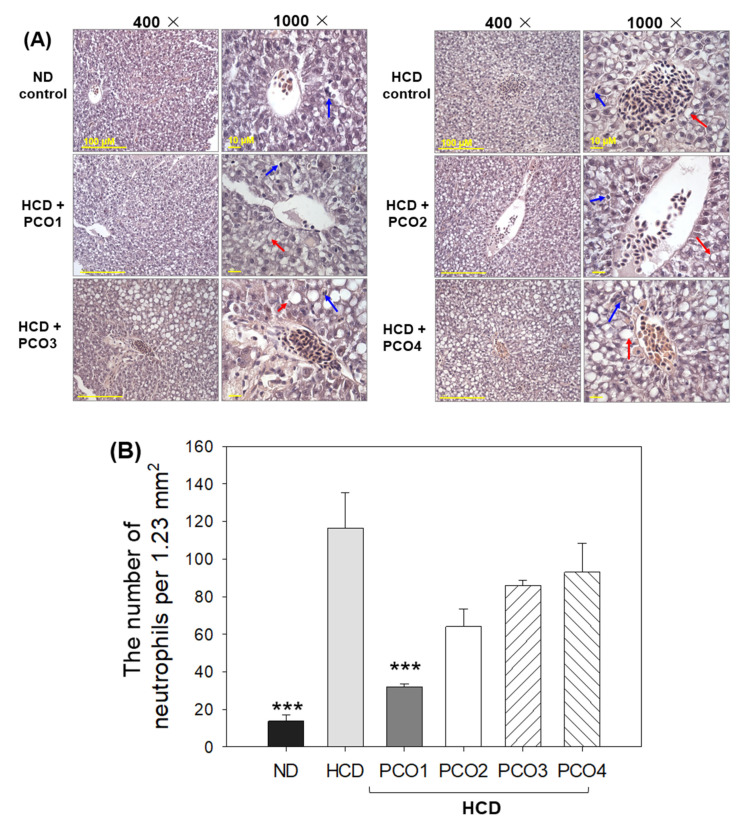
Observation of neutrophil infiltration in hepatic tissue from zebrafish after 12 weeks of supplementation of each policosanol (PCO) under high-cholesterol diet consumption. HCD, high-cholesterol diet; ND, normal diet; PCO1, Raydel policosanol; PCO2, Xi’an Natural policosanol; PCO3, Xi’an Realin policosanol; PCO4, Shaanxi policosanol. (**A**) Photographs showing the infiltration of neutrophils (blue arrow) and lipid droplets (red arrow) by hematoxylin & eosin (H&E) staining at 400× and 1000× magnification. Scale bar for 400× images (100 μm) and for 1000× magnified images (10 μm). (**B**) Determination of numbers in infiltrated neutrophils from the H&E staining using Image J software (http://rsb.info.nih.gov/ij/, accessed on 16 May 2023). The number of neutrophils (stained dark violet to blue color) of each group per designated area (1.23 mm^2^) was counted across five sections with every five views to obtain a semi-quantitative estimation of neutrophil infiltration. The statistical significance of the groups was indicated as *p* values at the top of the graph. Data are expressed as mean ± SEM. Statistical differences of multiple groups were compared using a one-way analysis of variance (ANOVA) with Dunnett’s post hoc test between the other group and the HCD group. ***, *p* < 0.001 versus HCD control.

**Figure 5 molecules-28-06609-f005:**
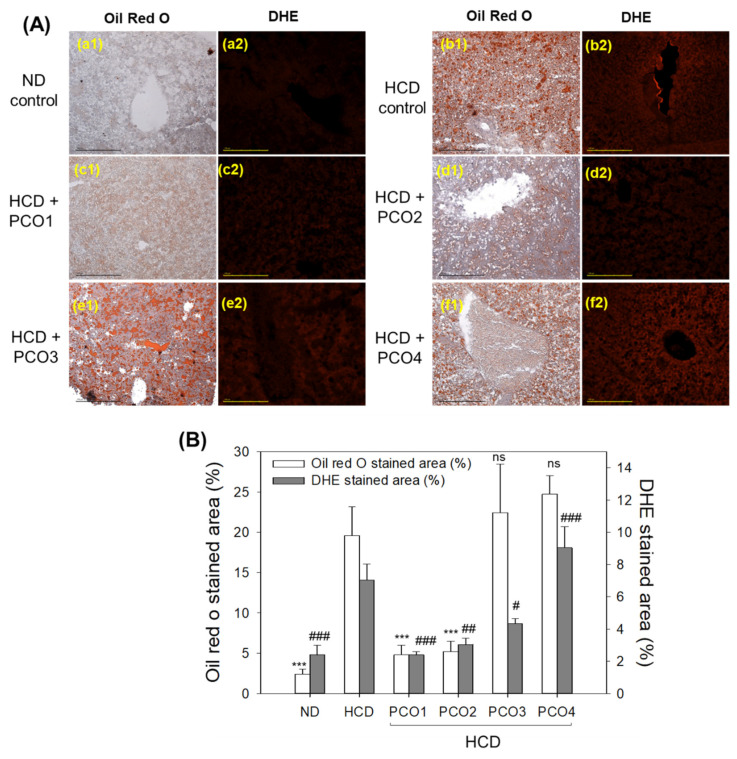
Comparisons of the fatty liver alteration and reactive oxygen species (ROS) examined by oil red O staining and dihydroethidium (DHE) staining, respectively, in zebrafish after 12 weeks of supplementation of each policosanol (PCO) under high-cholesterol diet consumption. HCD, high-cholesterol diet; ND, normal diet; PCO1, Raydel policosanol; PCO2, Xi’an Natural policosanol; PCO3, Xi’an Realin policosanol; PCO4, Shaanxi policosanol. (**A**) Exemplary images of oil red O stained (**a1**–**f1**) and DHE-stained (**a2**–**f2**) hepatic tissue of zebrafish sacrificed after 12 weeks’ consumption [scale bar = 100 μm]. (**B**) Assessment of the oil red O intensity and DHE fluorescence (Ex = 585 nm, Em = 615 nm) intensity employing Image J software (http://rsb.info.nih.gov/ij/, accessed on 16 May 2023). ***, *p* < 0.001 versus HCD from the oil red O stained area; Data are expressed as mean ± SEM. Statistical differences of multiple groups were compared using a one-way analysis of variance (ANOVA) with Dunnett’s post hoc test between the other group and the HCD group. #, *p* < 0.05 versus HCD from the DHE-stained area; ##, *p* < 0.01 versus HCD from the DHE-stained area; ###, *p* < 0.001 versus HCD from the DHE-stained area ns, not significant.

**Figure 6 molecules-28-06609-f006:**
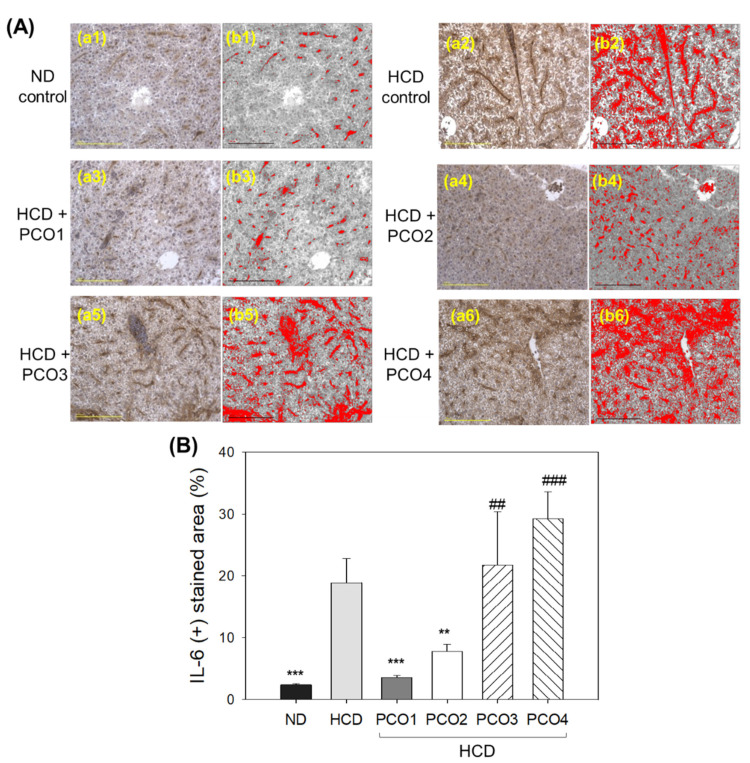
Comparison of the interleukin (IL)-6-stained area of immunohistochemistry (IHC) with hepatic tissue from zebrafish after 12 weeks’ supplementation of each policosanol (PCO) under high-cholesterol diet consumption. HCD, high-cholesterol diet; ND, normal diet; PCO1, Raydel policosanol; PCO2, Xi’an Natural policosanol; PCO3, Xi’an Realin policosanol; PCO4, Shaanxi policosanol. (**A**) Exemplary images (**a1**–**a6**) of IL-6-stained hepatic tissue after 12 weeks of consumption. The images (**b1**–**b6**) depict the IL-6 stained brown color, interchanged with red color [at brown color threshold value of lower limit (20) and upper limit (120)] to intensify the visualization of the IL-6-stained area using Image J software (http://rsb.info.nih.gov/ij/, accessed on 16 May 2023). The scale bar indicates 100 μm. (**B**) Assessment of the IL-6-stained area using Image J software (http://rsb.info.nih.gov/ij/, accessed on 16 May 2023). Data are expressed as mean ± SEM. Statistical differences in multiple groups were compared using a one-way analysis of variance (ANOVA) with Dunnett’s post hoc test between the other group and the HCD group. **, *p* < 0.01 versus HCD; ***, *p* < 0.001 versus HCD; ##, *p* < 0.01 versus HCD; ###, *p* < 0.001 versus HCD.

**Figure 7 molecules-28-06609-f007:**
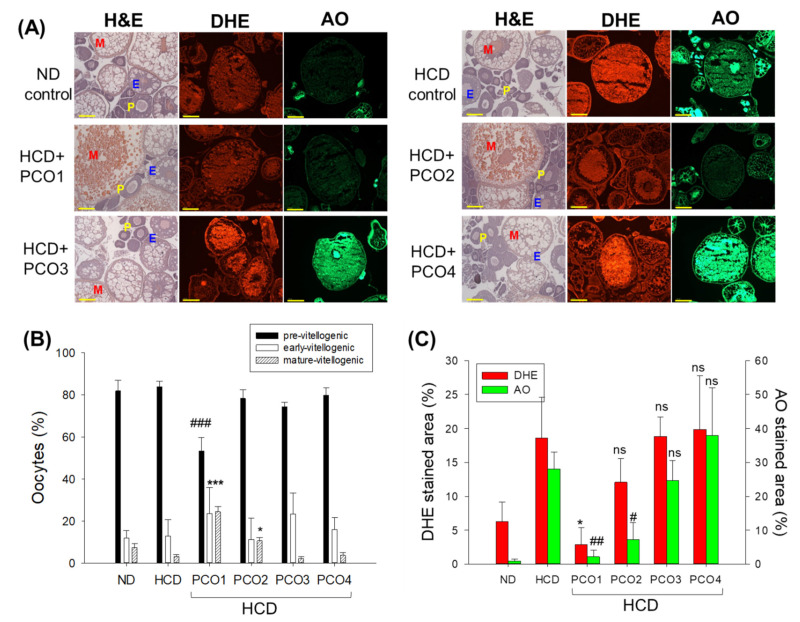
Comparisons of the ovarian cell morphology according to hematoxylin and eosin (H&E) staining, extent of reactive oxygen species (ROS) via dihydroethidium (DHE) staining, and the extent of apoptosis via acridine orange (AO) staining in zebrafish after 12 weeks’ supplementation of each policosanol (PCO) under high-cholesterol diet consumption. HCD, high-cholesterol diet; ND, normal diet; PCO1, Raydel policosanol; PCO2, Xi’an Natural policosanol; PCO3, Xi’an Realin policosanol; PCO4, Shaanxi policosanol. Data are documented as mean ± SEM. Statistical differences in multiple groups were compared using a one-way analysis of variance (ANOVA) with Dunnett’s post hoc test between the other group and the HCD group. (**A**) Representative images of H&E-stained, DHE-stained, and AO-stained ovarian cells. The scale bar indicates 100 μm. E, early vitellogenic oocytes; M, mature vitellogenic oocytes; P, pre-vitellogenic oocytes. (**B**) The percentage distribution of the stages in oocytes depends on the developmental stage. ***, *p* < 0.001 versus HCD in early and mature vitellogenic; ###, *p* < 0.01 versus HCD in pre-vitellogenic; *, *p* < 0.05 versus HCD in mature-vitellogenic. (**C**) Quantification of the DHE fluorescence (Ex = 585 nm, Em = 615 nm) intensity and acridine orange fluorescence in oocytes (Ex = 505 nm, Em = 535 nm) using Image J software (http://rsb.info.nih.gov/ij/, accessed on 16 May 2023). *, *p* < 0.05 versus HCD from the DHE-stained area; ##, *p* < 0.01 versus HCD from AO-stained area; #, *p* < 0.05 versus HCD from the AO-stained area; ns, not significant.

**Figure 8 molecules-28-06609-f008:**
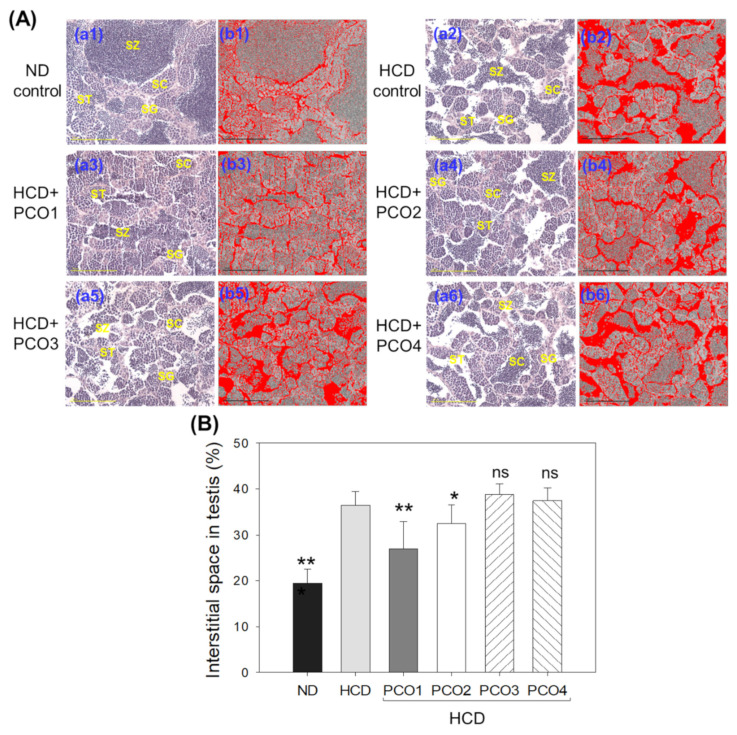
Comparison of the testicular histology in zebrafish after 12 weeks’ supplementation of each policosanol (PCO) under high-cholesterol diet consumption. HCD, high-cholesterol diet; ND, normal diet; PCO1, Raydel policosanol; PCO2, Xi’an Natural policosanol; PCO3, Xi’an Realin policosanol; PCO4, Shaanxi policosanol. (**A**) Representative images (**a1**–**a6**) of an H&E-stained testicular section. The images (**b1**–**b6**) depict the interstitial spaces in seminiferous tubules’ void areas (white color), interchanged with red color [at threshold values of lower limit (220) and upper limit (255)] to intensify the visualization of the interstitial spaces using Image J software (http://rsb.info.nih.gov/ij/, accessed on 16 May 2023). The scale bar indicates 100 μm. SC, spermatocytes; SG, spermatogonia; ST, spermatid; SZ, spermatozoa. (**B**) Quantification of the red area to compare the interstitial space in the testis using Image J software (http://rsb.info.nih.gov/ij/, accessed on 16 May 2023). Data are presented as mean ± SEM. Statistical differences of multiple groups were compared using a one-way analysis of variance (ANOVA) with Dunnett’s post hoc test between the other group and the HCD group. *, *p* < 0.05 versus HCD; **, *p* < 0.01 versus HCD; ns, not significant.

**Figure 9 molecules-28-06609-f009:**
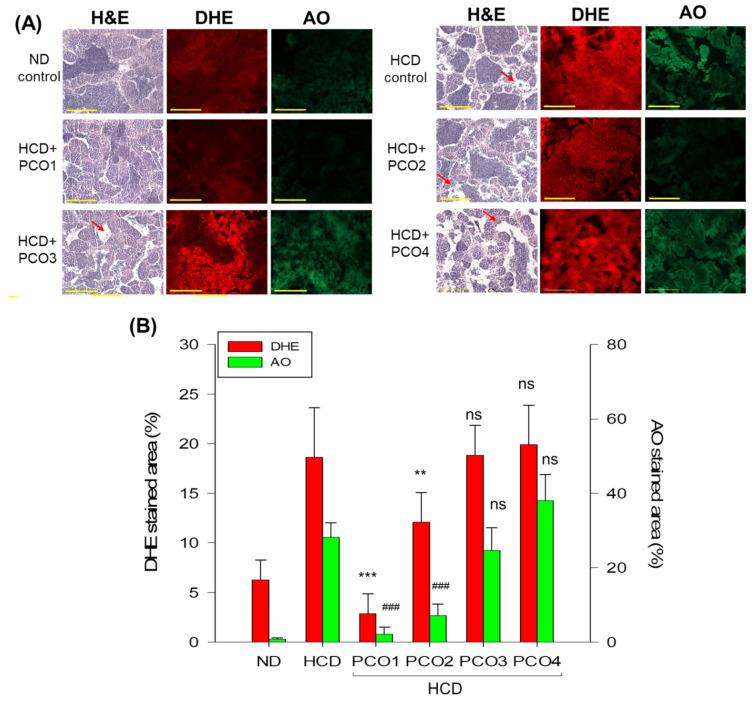
Comparison of the testicular cell morphology via hematoxylin and eosin (H&E) staining, the extent of reactive oxygen species (ROS) via dihydroethidium (DHE) staining, and the extent of apoptosis via acridine orange (AO) staining in zebrafish after 12 weeks’ supplementation of each policosanol (PCO) under a high-cholesterol diet. HCD, high-cholesterol diet; ND, normal diet; PCO1, Raydel policosanol; PCO2, Xi’an Natural policosanol; PCO3, Xi’an Realin policosanol; PCO4, Shaanxi policosanol. (**A**) Representative image of H&E-stained, DHE-stained, and AO-stained images of a testicular cell. The red arrow indicates increased interstitial area (scale bar = 100 μm). (**B**) Quantification of the DHE fluorescence intensity (Ex = 585 nm, Em = 615 nm) and AO fluorescence intensity (Ex = 505 nm, Em = 535 nm) using Image J software (http://rsb.info.nih.gov/ij/, accessed on 16 May 2023). Data are presented as mean ± SEM. Statistical differences of multiple groups were compared using a one-way analysis of variance (ANOVA) with Dunnett’s post hoc test between the other group and the HCD group. **, *p* < 0.01 versus HCD from DHE-stained area; ***, *p* < 0.001 versus HCD from DHE-stained area; ###, *p* < 0.001 versus HCD from the AO-stained area; ns, not significant.

**Figure 10 molecules-28-06609-f010:**
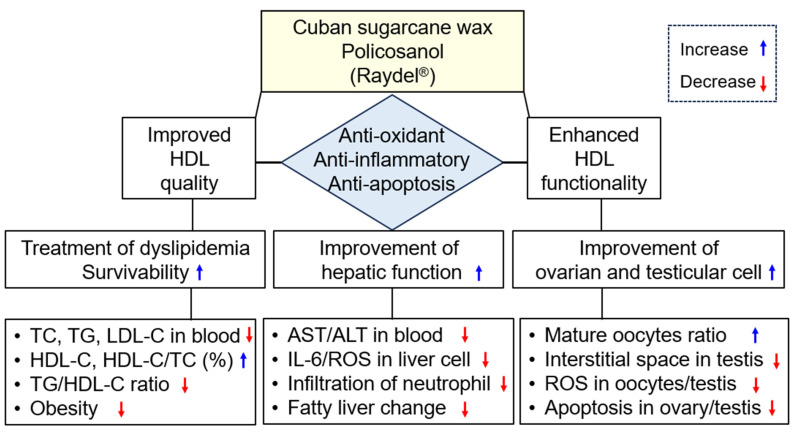
The beneficial effect of Cuban policosanol (Raydel^®^) in blood, liver, ovary, and testis after 12 weeks’ supplementation in hyperlipidemic zebrafish. AST, aspartate aminotransferase; ALT, alanine aminotransferase; HDL, high-density lipoproteins; IL-6, interleukin-6; ROS, reactive oxygen species; TC, total cholesterol; TG, triglyceride.

**Table 1 molecules-28-06609-t001:** Diet composition during 12 weeks of consumption of policosanol under a high-cholesterol diet (HCD).

		ND	HCD	HCD	HCD	HCD	HCD
	Groups	Control (n = 70)	Control (n = 70)	PCO1 Raydel Sugarcane Wax Alcohol (n = 70)	PCO2 Xi’an Natural Sugarcane (n = 70)	PCO3 Xi’an Realin Sugarcane (n = 70)	PCO4 Shaanxi Rice Bran (n = 70)
Diet composition (%)	Tetrabits ^1^	100	96	95.9	95.9	95.9	95.9
	Cholesterol (%, wt/wt)	-	4	4	4	4	4
	PCO (%, wt/wt)	-	-	0.1	0.1	0.1	0.1
	Octacosanol in PCO (mg/g) ^2^	-	-	692	356	69	492
	Final total PCO amount (mg) ^2^			982	610	592	518

^1^ Tetrabits, a brand name of zebrafish diet, was purchased from TetrabitGmbh (Melle, Germany). It contained 47.5% crude protein, 6.5% crude fat, 2.0% crude fiber, 10.5% crude ash, and vitamin A (29,770 IU/kg), vitamin D3 (1860 IU/kg), vitamin E (200 mg/kg), and vitamin C (137 mg/kg). ^2^ Adopted from reference [20]. ND, normal diet; HCD, high-cholesterol diet; PCO, policosanol.

## Data Availability

The data used to support the findings of this study are available from the corresponding author upon reasonable request.

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
