# Peer review of "Cuban Policosanol (Raydel®) Potently Protects the Liver, Ovary, and Testis with an Improvement in Dyslipidemia in Hyperlipidemic Zebrafish: A Comparative Study with Three Chinese Policosanols"

_molecules, 2023, doi:10.3390/molecules28186609_

Round 1

Reviewer 1 Report

The present study compared effects of four policosanols on dyslipidemia using zebrafish models via lipid analysis, histopathological analysis, histological analysis. The conclusion “12 weeks of Raydel policosanol consumption resulted in remarkable improvement of the blood lipid profile, amelioration of inflammation in the liver, and enhancement of the cell morphology in the ovary and testis with the highest survivability under the HCD”  is not fully supported by the present study, therefore, the conclusion should be based on the experimental results, and be rewritten.

1.      The manuscript title is too long. It should be concise.

2.      In the Abstract, rewrite “Although many policosanols from different sources, such as sugar cane, rice bran, and insects, have been marketed worldwide, there has been no comparative study of the in vivo efficacy using a hyperlipidemic animal model”. The authors did not mention what effects or properties policosanols possess, so it is confusing for readers why hyperlipidemic animal models are employed for the present study although various sources of policosanols were listed.

3.      Page 1, “A higher HDL-C quantity and improved HDL quality have been associated with …” , full names should be provided for the first mentioned abbreviations in the main text, such as HDL-C and HDL.

4.      There are too many grammatical errors in the whole manuscript to be pointed out one by one, such as “Zebrafish around 10 weeks old” and “2 μL blood” on page 20, “5 μm thick” and “7 μm tissue” on page 21. Including but not limited to the mentioned errors, correct them.

5.      Table 1describes diet composition. However, Table 1 is placed in Results on page 2, “suggesting that 4% cholesterol supplementation induced more acute death via progression of hyperlipidemia and hyperinflammation (Table 1).”. Table 1 does not support the conclusion.

 There are too many grammatical errors in the whole manuscript to be pointed out one by one, such as “Zebrafish around 10 weeks old” and “2 μL blood” on page 20, “5 μm thick” and “7 μm tissue” on page 21. Including but not limited to the mentioned errors, correct them.

Author Response

Thank you for your valuable comments and suggestions. We highly appreciate your kind efforts for giving us an opportunity to make scientifically and logically perfect manuscript. We tried to rectify the missing points raised by reviewers. We have modified the manuscript accordingly, and detailed corrections based on the comments of the reviewers are listed below point by point: 

Please find attached doc for the response and corrections

Reviewer 2 Report

Summary

The study comprehensively investigated the protective effects of Chinese and Cuban policosanols in hyperlipidemic zebrafish. Substantial evidence supports the positive impact of PCO1, which is observed in increased survivability, lowered levels of cholesterol and triglycerides, as well as reduced damage to liver, ovary, and testis tissues. However, the manuscript has many significant issues pertaining to imaging analysis, results interpretations and potential self-plagiarism. Furthermore, a number of minor flaws have been identified, which can be rectified to enhance the overall quality of the manuscript. These issues are outlined as follows:

Major:

Point 1: The similarity report indicates a 38% similarity, which is relatively high for a research article. Instances of self-plagiarism in the method and results sections were identified, accounting for upto 18% of the author's work. It is kindly advised to aim for an overall similarity score below 20% to ensure adherence to proper academic standards.

1.1 It appears that certain sections of the text are dislocated or sourced from other works. For example, in Figure 8 and 9, the reference to "180 min post-injection mean" raises confusion. As far as my comprehension goes, the administration of policosonols involved ingestion rather than injection, and there is no prior reference to injection in the manuscript. Notably, the authors have previously undertaken research involving microinjection treatment in zebrafish. Evidently, there is a case where the authors duplicated their text from a different work, and regrettably, they might not have been sufficiently diligent in eliminating the unrelated content.

1.2 The legend for Figure 9B describes the content as "Oil red O and DHE," which does not align with the actual content of the figure, namely the IL-6 graph.

Point 2: The assessment of neutrophil infiltration results appears to be inaccurate. Employing hematoxylin staining for nuclei does not provide a clear demarcation between native liver cells like hepatocytes and epithelial cells, and the infiltration of neutrophils. This limitation is evident in Figure A(b), where nuclei from the epithelium of the duct are also captured in the image analysis. It is crucial for the authors to elucidate their rationale for the study, the methodology employed, and their interpretation of the results. Clarity and justification are essential in this regard.

Point 3: In Figure 9, although it's evident that a high-cholesterol diet (HCD) leads to an increase in IL-6 within hepatic tissue, the quantification of IL-6 levels based solely on high-intensity areas appears to be unreliable. The authors should find alternative methods for image analysis.

3.1: Could the shift in color from grayish to brownish in hepatocytes signify an elevation in IL-6, or is the variation in the shade of brown merely an artifact? Moreover, why were these areas excluded from the image analysis? Clarification on these points is required.

3.2: In presenting the results, it might be beneficial to describe the findings in terms of areas with high IL-6 staining. There appears to be a discernible pattern here. Could this pattern be attributed to specific locations, such as blood vessels or bile canaliculi, or is there another underlying factor? Providing more context on this matter would enhance the understanding of the observed patterns and justify the results.

Point 4: An essential improvement lies in the combination of figures 2 to 5, followed by a revision of the results section. Instead of providing individual numerical values for each bar in the graphs, it is advisable to present the results in a more concise and pointed manner. To enhance clarity and focus on the core concepts of the manuscript, it's suggested to exhibit graphs solely related to primary themes such as Total Cholesterol and standard ratios (HDL/TC, LDL/HDL), while relocating graphs of lesser significance to the supplementary materials. The present structure hampers the central message's comprehension, and there's a sense that the new graphs merely replicate prior data in different forms. This modification would significantly streamline the information presentation and reinforce the manuscript's key points.

Point 5: While the stated objective was to evaluate the effectiveness of various policosanol types, the subsequent discussion predominantly centers around the favorable outcomes associated with PCO1. To rectify this, it is strongly recommended that a detailed examination of the varying constituents present in the four PCOs be incorporated into the discussion.

Minors

Introduction

Paragraph 1, the last sentence. It is confusing how low HDL-C was given as example of dyslipidemia and diabetes. Please re-write the sentence.

Method

1) Please provide additional details regarding the image normalization process, specifically the methodology employed to calibrate arbitrary color and fluorescent intensity values across images. Furthermore, kindly elaborate on the specific functions utilized within the ImageJ tool. It is of particular interest to understand the procedure employed for the identification and selection of the areas highlighted in red within the images.

Results

1) Figures require unity and improved formatting. There is a lack of consistency in text size and placement. Repetition has been identified in subfigure titles and axes. Kindly focus on these details.

2) The authors utilized the term 'conversion of red intensity' for all image analysis, but it does not convey the accurate meaning. Please provide specific terminology for each individual analysis.

3) Please arrange the order of the results section correctly (there is no 2.7 and 2.8 BUT 2.9?)

4) Correct the language in all titles that begin with 'Histological Analysis...' for both the results and methods sections.

Note: None of the comments were intended to discourage the author in any way. It is evident that you have already put a significant amount of effort into conducting the experiments, which is reflected in the manuscript. I kindly urge you to invest more effort in data analysis, result presentation, and writing.

As suggested in the previous section. Also, a language editing service could help with self-plagiarism problems.

Author Response

Thank you for your valuable comments and suggestions. We highly appreciate your kind efforts for giving us an opportunity to make scientifically and logically perfect manuscript. We tried to rectify the missing points raised by reviewers. We have modified the manuscript accordingly, and detailed corrections based on the comments of the reviewers are listed below point by point: 

Please find attached doc for response and corrections

Round 2

Reviewer 1 Report

The manuscript has been well modified for the publication.

Author Response

Thank you very much for acceptance.

We really appreciate your comments.  

Reviewer 2 Report

The author addressed most of the pointed issues; however, a few corrections are still required as follows:

1) In the previous round of review, an issue was raised regarding the identification of neutrophil infiltration. The author has re-analyzed the results and provided supporting references. However, Figure 4a does not adequately represent the findings and raises concerns about the methodology used for Figure 4b. While H&E staining can be employed to identify neutrophils or mononuclear cell infiltration in tissues, the morphology and area of the nuclei associated with these cells are crucial, as demonstrated in the provided references. In the case of hepatic tissue, it is expected that the author would focus on areas of blood vessels such as the central vein, portal vein, or portal artery and use the same site for comparison between the experimental groups. However, some of the subfigures appear to concentrate on the bile duct. Describe how the neutrophil was identified, especially in the represented images.

2) Please also add the scale bar in Figure 4a.

3) In the previous round, the authors were advised to address the issue with the term 'conversion of red intensity' used for all image analysis. This term was deemed inaccurate, especially considering that the images were converted to grayscale, eliminating red intensity. Instead, the red areas represent specific regions for analysis. The author did not adequately address this issue. It is essential to find the appropriate terminology to label the figures and clarify what the red areas represent in each figure.

In the previous round, there was a request to 'Correct the language in all titles that begin with 'Histological Analysis...' for both the results and methods sections.' The author corrected some of the titles but failed to fix all.

Author Response

Thank you for your valuable comments and suggestions. 

Please find detail point-to point response
